# Logistics Village Location with Capacity Planning Problem, an MILP Model Approach

**Amirhossein Baghestani** [1], **Mohammadhossein Abbasi** [2], **Saeed Rastegar** [3], **Amir Reza Mamdoohi** [2,*], **Atoosa Afaghpoor** [4] **and Mahmoud Saffarzadeh** [2]

1   Faculty of Civil, Water and Environmental Engineering, Shahid Beheshti University, Tehran 1983969411, Iran
2   Faculty of Civil & Environmental Engineering, Tarbiat Modares University, Tehran 14115, Iran
3   Industrial Engineering School, Iran University of Science and Technology (IUST), Tehran 1684613114, Iran
4   Faculty of Urban Development, Tehran University, Tehran 141556458, Iran
*   Correspondence: armamdoohi@modares.ac.ir; Tel.: +98-21-8288-4925

**Abstract:** The optimal location for establishing logistics centers is of great importance in reducing logistics costs and improving supply chain efficiency. This paper aims to provide a conceptual framework for finding the optimal location and capacity for a logistics village establishment using mixed-integer linear programming (MILP). The proposed model is applied on Qazvin province, Iran, as a developing country with a strategic location in international transport corridors. Unlike previous research, the proposed approach considers various logistics operations such as warehousing, refrigeration, sorting, and packaging, along with their capacities as distinct decision variables. The study area is divided into 6972 blocks of $1.5 \times 1.5$ km, of which 59% are infeasible and excluded due to environmental and natural hazard constraints. The MILP model is then applied in the GAMS for each feasible block to identify the best alternatives for the logistic village establishment with maximum total profit. Based on the results, total freight imported to Qazvin province is directly transferred to their final destinations without visiting the logistics village, while around 98% of exports of Qazvin province would first enter the logistics village to get a service before delivering to customers.

**Keywords:** logistics village; hub location problem; mixed-integer linear programming; developing country

## 1. Introduction

Transportation and logistics are significant contributors to economic growth. Logistics systems can support trading markets, improve freight transportation, facilitate the development of trade exchanges, and ensure that the right customers are served at the right time [1,2]. As a component of logistics systems, logistics centers can be configured to fulfill a variety of functions, such as storage, assembly, transport, distribution, and delivery [3]. A logistics center is the hub of activity relating to transport, storage, and distribution, connecting different transportation modes and performing broad logistic functions [4]. Using logistics centers in transportation networks significantly reduces transportation costs and the number of connection routes, energy consumption, and value-added services [5,6]. Due to their benefits in the last few decades, logistics centers have emerged as essential components in supply chains and international transportation systems [7]. These centers are classified based on their location, functionality, and value-added services, including warehouses, freight villages, and main port terminals [7,8].

Due to the continuing importance of emerging markets, both the public and private sectors are increasingly investing in logistics centers. Large logistics service providers currently serve more than 100 countries, and the most extensive express companies cover around 200 countries [9]. The shift from state-owned enterprises to private companies, as in the USA, represents an important step towards developing market structures. Some of the current examples include the freight villages of Alliance Texas [10], the Raritan

Center [11], and the Virginia inland port [12]. European countries have also conducted extensive investigations on the development of logistics centers, leading to each country having a built or under construction type of logistics center [13,14]. Interporto Bologna, GVZ Bremen, and DIRFT Logistics Park are some examples of European logistics centers. The inclusion of the least developed countries and new trade corridors is supposed to meet the demands of new customers and market entrants. In recent years, Asian countries have extensively studied the development of logistics infrastructures [15–18].

One of the most strategic decisions in logistics is choosing the logistics center's site. It is crucial to locate logistics facilities in areas where trade flow connects consumption and production patterns [19]. The importance of logistics centers in the management of freight transportation networks has always prompted researchers to study their location and design. In site selection, many factors should be considered to deal with conflicting goals that make the problem challenging. Indeed, the selected location should maximize both the quantity and quality of the trade flow while considering related costs, resource limitations, and risks [20]. Therefore, an extensive body of the logistics literature has focused on location problems to satisfy the demand and reduce transport and infrastructure costs [21].

Different aspects are considered for locating logistics centers that are used as criteria or constraints in modelling. According to Vieira and Luna [22], these aspects can be categorized as: (1) transport, which is concerned with transportation costs, time or distance travelled; (2) hub functionality, which is involved with capacity, activities undertaken in the hub, expenses, operational cost, etc.; (3) investment, which is required for facilities' construction; (4) demand and supply; (5) market, which deals with the proximity to customers and the possibility of expanding the coverage; (6) policy, which includes policy developments, existing legislation, and the advantages of tax incentives; and (7) environment, which is related to terrain profiles, environmental protection, and geography. A review of previous research in hub location problem (HLP) indicates that multi-criteria decision making (MCDM) techniques and linear integer programming (LIP) are of great interest among researchers. Besides MCDM advantages, the main limitation of this method is that the weight of each criterion is subjective and is based on decision-maker preferences which affects the accuracy of the results [23].

Using LIP and genetic algorithms, Xin et al. [24] studied the HLP in Beijing, China, by minimizing transportation costs. Çakmak et al. [25] located logistics hubs in Istanbul by integrating GIS and mathematical optimization algorithms. They presented an optimization model to select the most suitable location for logistics hub locations using meta-heuristic algorithms. Logistics hubs are generally defined as linking points—infrastructure facilities and nodal points [26]. After reviewing the criteria from the perspective of stakeholders, Nguyen and Notteboom [27] used MCDM method to solve HLP for a dry port in Vietnam. Aksoy and Gursoy [28] ranked the logistics villages in terms of establishment priority in Turkey, either already established or in the planning/establishment phase. Employing AHP, the weight of different criteria was calculated based on 17 experts' opinions and ranking was conducted using ELECTRE method. It should be noted that the criteria considered were supply and demand, social and infrastructure facilities, cost, environment, and location. Additionally, Hanifha et al. [29] located a logistics company in India using the gravity model and mixed-integer linear programming (MILP). Using the gravity model, they identified potential areas based on the distance from the main company, and using MILP, they sought a suitable location to construct new facilities considering total cost minimization. Aiming to minimize total costs, including transportation and construction costs, Wang et al. [30] used genetic algorithm with two layers of chromosome coding to select the optimal location of logistics hub for agricultural products. Furthermore, Yazdani et al. [31] presented a two-stage integrated decision model for locating logistics centers in Spain. One of the modeling features of this research is to consider preferential conflicts and uncertainty of information in the modeling process.

According to the previous studies, it can be concluded that locating logistics centers is so critical in integration of freight network and consequential benefits. Although Iran has a

strategic location in international transport corridors with a favorable growth in logistics performance index (LPI) in 2018 compared to previous years, the country ranked 64th among 167 countries [32]. Figure 1 shows Iran's logistics performance compared to UAE (11th) as the top performer in Asia and Germany (1st) as the top performer in the world in terms of various logistics dimensions, such as customs, infrastructure, international shipments, logistics competence, tracking and tracing, and timeliness. Obviously, there is a significant difference between Iran and these two countries, highlighting the need for paying more attention to this sector.

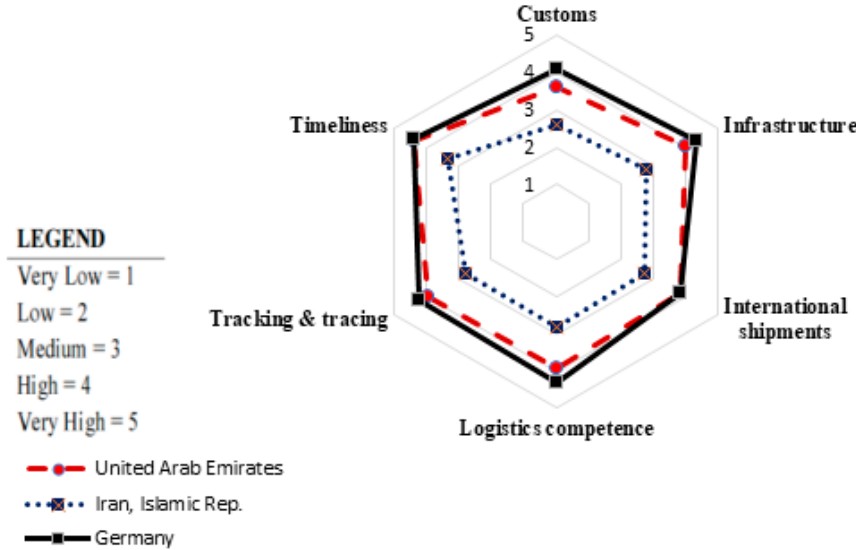

**Figure 1.** Iran logistics performance index compared to UAE and Germany [32].

The current study aims at proposing a mathematical model to solve a logistics center location selection problem for a real-sized case of Qazvin, Iran. In this regard, a mixed-integer linear programming (MILP) approach including novel decision variables, objectives, and constraints is proposed. To the best of our knowledge, previous studies on logistics center location problems did not consider various logistics operations such as warehousing, refrigeration, sorting and packaging. Hence, previous hub models considered capacity as a decision variable, taking into account the total loading capacity of the hub [33]. However, the proposed model takes into account these services and considers their capacities as distinct decision variables of the model. Furthermore, in contrast to most previous studies, the proposed approach includes investment, operations, and maintenance costs (as annualized costs over the lifetime of the facilities) as well as the revenues generated by providing the aforementioned services. This approach enables the decision-maker tradeoff between development costs and operational revenues. In other words, capacity variables are used to estimate the total area needed (and consequently development costs) for each facility using capacity-to-area coefficients. This coefficient lets the model simultaneously determine the logistics village's optimal location and capacity (considering its various operations).

The rest of this paper is organized as follows. Section 2 describes the study area and Section 3 presents the methodology, including the mathematical model. Section 4 shows the results along with discussion. The paper concludes with a summary of findings and outlines future research directions.

## 2. Study Area

Freight transportation and logistics in Iran provided considerable opportunities and benefits due to its strategic location, i.e., at the crossroads of Middle East trade links with Central Asia and the Arab states, as well as easy access to EU transit routes and international waters. However, the World Bank report indicates the potential of this sector has not been fully utilized to date, highlighting the need for more attention to logistics

center developments [34,35]. The current study focuses on Qazvin province, located in the northern half of Iran with an area of 15,626 km², occupying less than one percent of the country's area.

Qazvin's location at the international North-South and East-West transport corridors, as well as the Silk Road (Figure 2) makes it a strategic transit point for large amounts of cargo. The Silk Road is a trade corridor connecting Asia and the Middle East with the Western world by passing through China, Kazakhstan, Tajikistan, Uzbekistan, Turkmenistan, Iran, Turkey, Bulgaria, and Germany. Another corridor is the Islamabad-Tehran-Istanbul Rail Corridor (ITI), allowing Pakistan to transit to European countries through Iran and Turkey. Additionally, Qazvin is located at the Iran-Turkey-Bulgaria-Hungary-Germany (ITBHG) corridor that connects Tehran to Europe. This could be considered as one of the reasons that more than 36.5% of Iran's transit cargo (about 4.5 million tons annually) passes through Qazvin province [36].

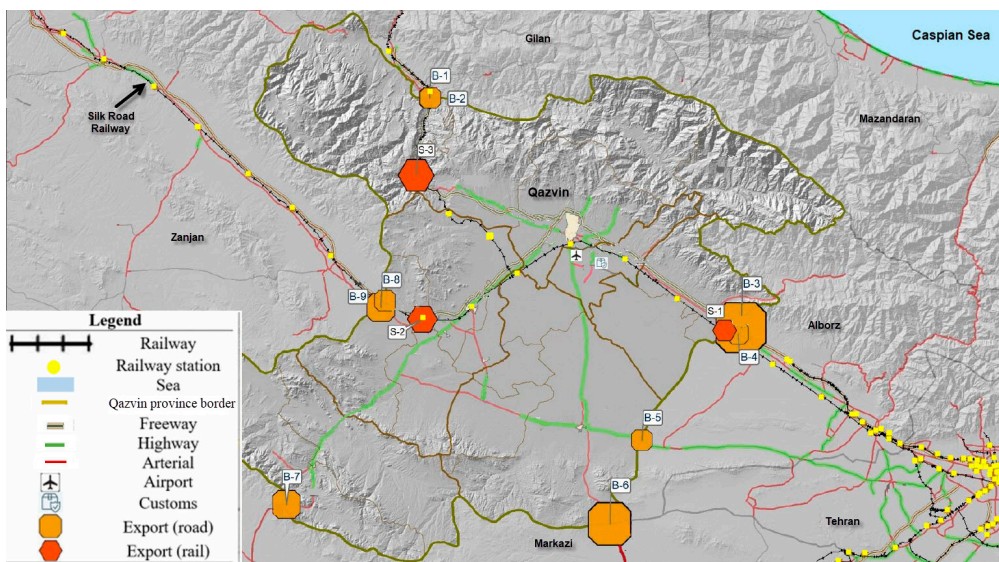

**Figure 2.** Geographical location and logistics infrastructure of Qazvin. B and S notations are indicating the border road and rail stations, respectively.

In 2018, Qazvin customs' import and export tonnages were around 121 million ($625 million equivalent) and 603 million tons ($558 million equivalent), respectively. The average weight of cargo exported through the province during the five-year period was about 133 million tons, so the trend of changes in the weight of cargo exported through the province has always been upward and has experienced a growth of 173 percent. Table 1 presents the tonnage of cargo export, import, and intra-province transits in Qazvin stratified by transportation modes including rail, marine, and road transport systems. It can be concluded that a significant amount of goods is transported by road due to the lack of integration in freight transportation in Iran, especially Qazvin. Further, a proportionality can be seen between the import and export of road transportation, while in the rail transportation, the import is significantly higher than the export.

**Table 1.** Import and export of Qazvin province.

| Mode | Export | | Import | | Total | |
|---|---|---|---|---|---|---|
| | Tonnage | Share (%) | Tonnage | Share (%) | Tonnage | Share (%) |
| Road | 7,809,626 | 94.45 | 7,820,107 | 97.63 | 15,629,733 | 96.02 |
| Rail | 58,094 | 0.70 | 189,267 | 2.37 | 247,361 | 1.52 |
| Marine | 400,664 | 4.85 | - | - | 400,664 | 2.46 |
| Total | 8,268,384 | 100 | 8,009,374 | 100 | 1,627,758 | 100 |

### 3. Method

#### 3.1. Problem Formulation and Assumptions

The objective of this study is to determine the optimal location and size (capacity) of a "logistics village". A logistics village is a special type of logistics center providing intermodal transportation, warehousing, refrigeration, sorting, and packaging as well as peripheral services such as vehicle maintenance services. Centralization of these services enables it to benefit from the economy of scale, hence, lowering the costs of different stakeholders. The logistics village links out of province origin/destination nodes to send/receive cargo to the supply/demand zones inside the province as well as enabling intra-province transportations. It also provides the possibility of transportation modal change and other value-adding services for transit cargo with both origins and destinations outside the province. A schematic view of the logistics village and its components is provided in Figure 3.

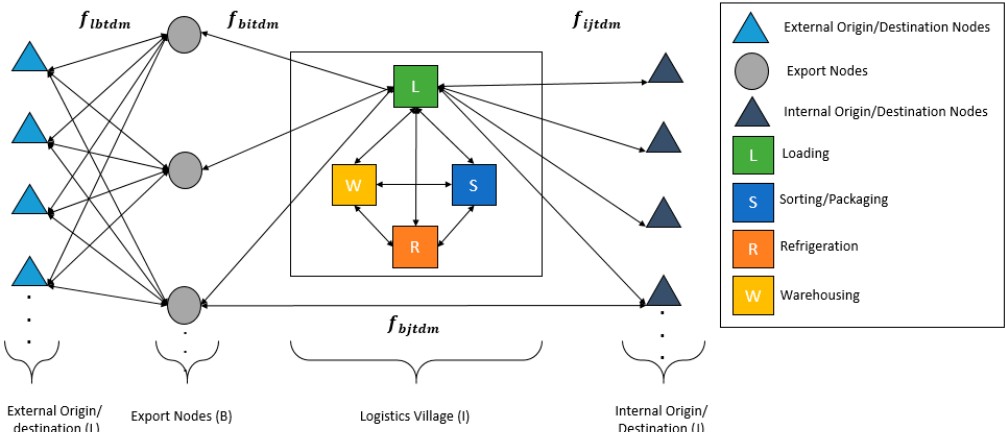

**Figure 3.** Configuration of the network and the logistics village.

The assumptions of the model are as follows:

- Multiple cargo types including agricultural, industrial, mineral, structural, alimentary, and bestial as well as various transportation modes are considered;
- Multiple value-adding services including warehousing, refrigeration, sorting, and packaging are assumed to be implementable in the logistics village;
- Decision variables of the model include logistics village location, cargo volume transported between inside and outside province nodes, transportation mode, capacity of warehousing, refrigeration, sorting, and packaging services;
- Total costs include lifetime costs of facility (road, rail, power, water, etc.) provision (including investment, operations, and maintenance costs), transportation costs and warehousing, refrigeration, sorting and packaging service costs. These costs are annualized over their lifetime period;
- Revenues include revenues generated by providing the aforementioned value-adding services. The model provides a trade-off between these revenues and costs and makes it possible to select a location that best satisfies various stakeholders of the logistics village (i.e., cargo owners, transportation companies, as well as the logistics village investors/owners);
- Qazvin province's imports and exports are estimated through the lifetime of the logistics village based on various growth models and the most precise forecasts (in terms of relative error in forecasting training data) are applied for each cargo type.

A schematic view of the hierarchy of cost and revenues for various stakeholders is provided in Figure 4. As indicated, total costs consist of fixed costs regarding the infrastructures and variable costs including transportation and value-added tariff and service costs. In terms of total revenues, two items are taken into account: revenues related

to the added value of the cargo (from services provided at logistic villages) and revenues related to value-added services.

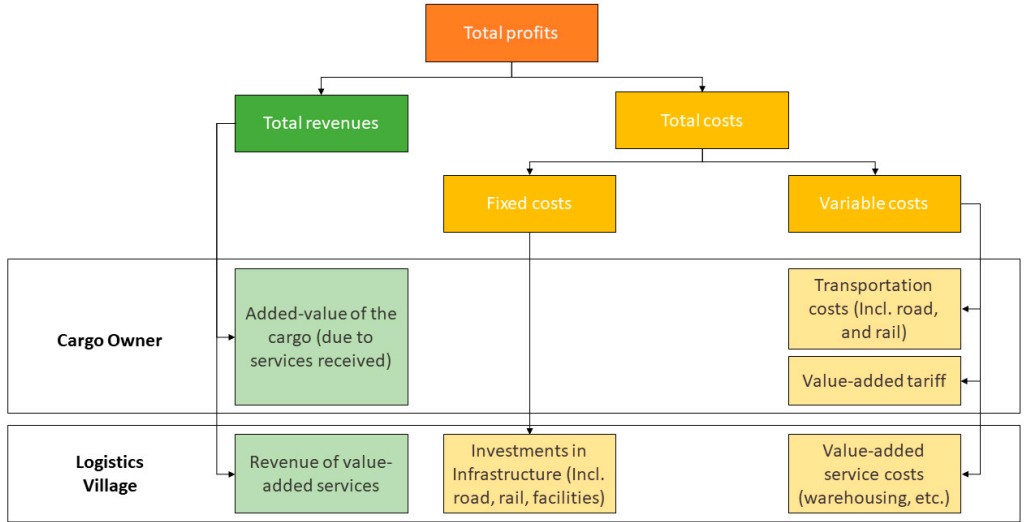

**Figure 4.** Hierarchy of revenues and costs of the logistics village by stakeholder.

### 3.2. Mathematical Model

A MILP model has been formulated using various variables, constraints, and parameters (Table 2) to find the optimal location of a logistics village location.

$$Maximize \sum_i \sum_t \sum_k V_{tk}y_{itk} - \sum_l \sum_b \sum_t \sum_d \sum_m D_{lbm}C_{tm}f_{lbtdm} - \sum_b \sum_i \sum_t \sum_d \sum_m D_{bim}C_{tm}f_{bitdm} - \sum_b \sum_j \sum_t \sum_d \sum_m D_{bjm}C_{tm}f_{bjtdm} - \sum_i \sum_j \sum_t \sum_d \sum_m D_{ijm}C_{tm}f_{ijtdm} - \sum_i \sum_e D_{ie}C'_{ie}x_i - \sum_i \sum_k C''_{k}z_{ik} \quad (1)$$

$$\sum_b \sum_m f_{lbtdm} = \sum_m F_{ltdm} \qquad \forall l,t,d \quad (2)$$

$$\sum_i f_{bitdm} + \sum_j f_{bjtdm} = \sum_l f_{lbtdm} \qquad \forall b,t,d,m \quad (3)$$

$$\sum_j \sum_m f_{ijtdm} = \sum_b \sum_m f_{bitdm} \qquad \forall i,t,d \quad (4)$$

$$\sum_i \sum_m f_{ijtdm} + \sum_b \sum_m f_{bjtdm} = \sum_m F_{jtdm} \qquad \forall j,t,d \quad (5)$$

$$y_{itk} \leq \sum_b \sum_m f_{bitdm} + \sum_j \sum_m f_{ijtdm} \qquad \forall i,t,k,d \neq 1 \quad (6)$$

$$z_{ik} \geq \sum_t A_k y_{ikt} \qquad \forall i,k \quad (7)$$

$$z_{ik} \leq M^b x_i \qquad \forall i,k \quad (8)$$

$$\sum_i x_i \geq M^b in^G \qquad \forall i \quad (9)$$

$$f_{ijtdm} \leq F_{jtdm}x_i \qquad \forall i,j,t,d,m \quad (10)$$

$$f_{bitdm} \leq F_{ltdm}x_i \qquad \forall i,b,t,d,m \quad (11)$$

$$x_i \in \{0,1\} \qquad \forall i \quad (12)$$

$$f_{lbtdm} \geq 0;\ f_{bitdm} \geq 0;\ f_{bjtdm} \geq 0;\ f_{ijtdm} \geq 0; \qquad \forall l,b,i,j,t,d,m \quad (13)$$

$$y_{itk} \geq 0 \qquad \forall i,t,k \quad (14)$$

$$z_{ik} \geq 0 \qquad \forall i,k \quad (15)$$

**Table 2.** Notations used in the model.

| Symbol | Definition |
|---|---|
| **Sets and Indices** | |
| $I$ | Set of feasible logistics village locations, indexed by $i \in I$. |
| $J$ | Set of all (feasible and infeasible) nodes located in the province indexed by $j \in J$. |
| $B$ | Set of provincial border nodes indexed by $b \in B$. |
| $L$ | Set of origin/destination nodes located out of province indexed by $l \in L$. |
| $T$ | Set of cargo types indexed by $t \in T$ (agricultural, industrial, mineral, structural, alimentary, and bestial). |
| $M$ | Set of transportation modes indexed by $m \in M$ (rail, road). |
| $D$ | Set of transport link directions indexed by $d \in D$ (from/to logistic village, from/to the province). |
| $E$ | Set of necessary facilities indexed by $e \in E$ (rail, road). |
| $K$ | Set of services provided by the logistics village indexed by $k \in K$ (warehousing, sorting, and packaging). |
| **Parameters** | |
| $D_{lbm}$ | Distance between external nodes ($l$) and border nodes ($b$) using transportation mode m. |
| $D_{bim}$ | Distance between border nodes ($b$) and feasible logistics village location node ($i$) using transportation mode m. |
| $D_{bjm}$ | Distance between border nodes ($b$) and all feasible and infeasible nodes ($j$) using transportation mode $m$. |
| $D_{ijm}$ | Distance between feasible logistics village location node ($i$) and all feasible and infeasible nodes ($j$) using transportation mode $m$. |
| $D_{ie}$ | Distance between feasible logistics village location node ($i$) and facility $e$. |
| $C_{tm}$ | Unit cost of transporting cargo type $t$ using transportation mode $m$ (IRR per ton-kilometer). |
| $C'_{ie}$ | Annualized unit cost of establishment, operation and maintenance of facility $e$ (IRR per kilometer). |
| $C''_{k}$ | Unit cost of capacity development for service $k$ (IRR per ton). |
| $F_{jtdm}$ | Demand/supply of all feasible and infeasible nodes ($j$) for cargo type $t$ in direction $d$ using transportation mode $m$. |
| $F_{ltdm}$ | Demand/supply of external nodes ($l$) for cargo type $t$ in direction d using transportation mode $m$. |
| $A_k$ | Coefficient of area needed for service $k$ (per m$^2$). |
| $V_{tk}$ | Revenue (value-added) by applying service $k$ on cargo type $t$ (IRR/ ton). |
| $Min^G$ | Minimum number of grids (area) of the logistics village. |
| $M^b$ | A sufficiently large number. |
| **Decision Variables** | |
| $x_i$ | Binary variable; $x_i = 1$ if feasible logistics village location node ($i$) is selected for logistics village; 0 otherwise. |
| $f_{lbtdm}$ | Volume of cargo type t transported from external nodes ($l$) to border nodes ($b$) using mode m in direction $d$. |
| $f_{bitdm}$ | Volume of cargo type t transported from border nodes ($b$) to feasible logistics village location node ($i$) using mode $m$ in direction $d$. |
| $f_{ijtdm}$ | Volume of cargo type $t$ transported from feasible logistics village location node ($i$) to all feasible and infeasible nodes ($j$) using mode m in direction $d$. |
| $f_{bjtdm}$ | Volume of cargo type $t$ transported from border nodes ($b$) to all feasible and infeasible nodes ($j$) using mode m in direction $d$. |
| $y_{itk}$ | Volume of cargo type $t$ receiving service type $k$ at feasible logistics village location node ($i$). |
| $z_{ik}$ | Capacity of service type $k$ at feasible logistics village location node ($i$). |

Equation (1) defines the objective function, i.e., profits of the logistics village. The profit is calculated by subtracting transportation, facility establishment, and service costs from the revenue earned by the value-added services provided in the logistics village. Equation (2) indicates that the total volume of cargo type t sent from out of province nodes need to be equal to total cargo received in the province, and vice versa. Equation (3) guarantees that total cargo type t received in border point b must be equal to total cargo sent from the same border point. According to Equation (4) total cargo type t sent and received by the logistics village must be equal. Equation (5) functions similarly for out of province destinations/origins. Based on Equation (6), total cargo receiving service at logistics village must be lower or equal to total cargo received at the village. Equation (7) limits total cargo receiving service k at logistics village to the capacity of logistics village for providing service k. This constraint helps the designers estimate total area needed for providing service type k. Equation (8) binds the capacity of service provision at each node to the selection of the node for logistics village. Equation (9) limits total blocks selected for the logistics village. Equations (10) and (11) bind sending and receiving cargo to/from each

potential node to its selection as part of the logistics village. Finally, Equations (12)–(15) indicate the type of variable $x_i$ as binary and variables $f_{lbtdm}, f_{bitdm}, f_{bjtdm}, f_{ijtdm}, y_{itk}$, and $z_{ik}$ to be non-negative real variables.

As the resultant model is NP-Hard, we propose a two-step approach to lower the time complexity of the problem, as follows.

1.  Pre-process the solution space and determine the feasible region using Geographic Information System (GIS) based overlaying method. This approach lowers the number of variables (in particular integer and binary variables) and constraints of the model;
2.  Solve the model using the resultant reduced solution space.

This approach can be applied as an alternative to or alongside cutting plane methods to reduce the time complexity of optimization models and make it possible calculate their exact and precise solutions in a reasonable time. It is noteworthy that the resultant model consists of 51,765 variables and 39,706 constraints. We applied CPLEX implementation of the branch and cut algorithm using GAMS 24.1.2 to solve the problem. The code is run using a 4-core 4.2 GHz Intel 1140G7 processor with 8 gigabytes of RAM.

**4. Results**

Research conceptual model, aiming at maximizing the total profit, is illustrated in Figure 5. According to this framework, the MILP model's input (Section 4.1) is used to determine the best alternatives among feasible divided blocks of 1.5 × 1.5 km. After determining the criteria affecting logistics village location, infeasible regions (Section 4.2) excluded and feasible areas will be studied. The model's output will be then discussed in Section 4.3.

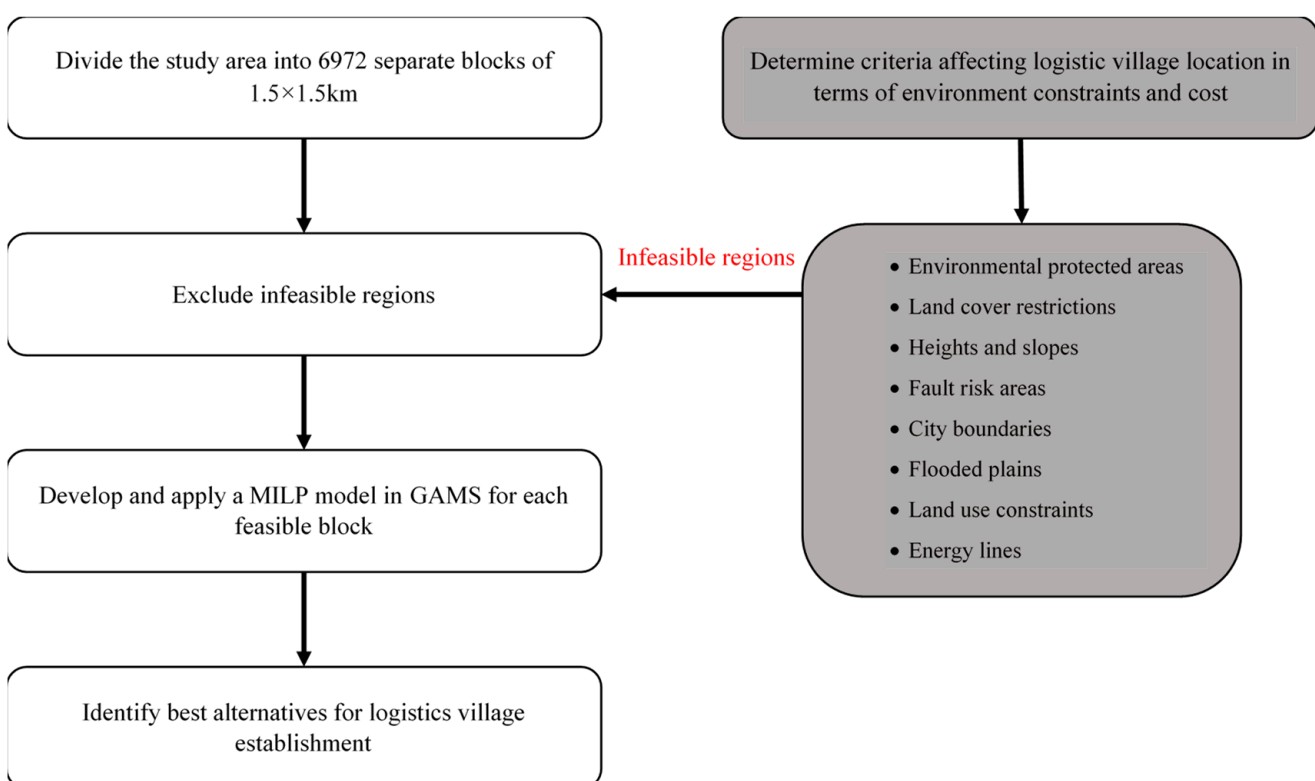

**Figure 5.** Flowchart of the solution approach.

*4.1. Model Inputs*

A single-hub location problem with multi-product allocation has been considered to find the most appropriate site for a logistics village in Qazvin. The objective function and constraints are formulated as a MILP in the GAMS software using the CPLEX solver. There

are two approaches to the space location problem, namely, continuous and discrete. In a continuous location problem, optimal results might not be practical due to some executive considerations. Therefore, the discrete approach has been used for hub location in this study. In this approach, the appropriate candidates for hub locations are selected and the best solutions are ranked based on the objective function value. In this regard, Qazvin province has been divided into 6972 separate blocks of 1.5 × 1.5 km. Six commodity groups have been considered in the model including agricultural, industrial, mineral, structural, alimentary, and bestial.

*4.2. Feasible Region*

Logistics centers should not be located in certain areas because of natural and unnatural constraints because of their strategic importance. It is, therefore, necessary to identify the feasible region by removing infeasible areas before running the proposed model. This study considers the following eight measures to identify infeasible areas. Next, the model will be run for the remaining blocks to provide a feasible solution. Infeasible areas considered in this research are as follows:

- Environmentally protected lands: These areas should be excluded from feasible blocks because of their recognized natural, ecological, or cultural values. Logistics activities might result in environmental pollutions which would be harmful to plant and animal species. Some examples of such lands include national parks, national natural landmarks, and wildlife shelters;
- Land cover constraints: Since the establishment of logistics centers in large land areas would require cutting trees, land cover constraints should be applied to the feasible regions. Wetland swamp, groves, shrubs, and forests are some cases that are not allowed for the establishment of a logistics center;
- Sloped lands: In such cases, excavation and embankment costs require more funding sources. Since logistics centers should be linked to rail network systems, technical constraints of railways should also be considered. Hence, lands with a slope of more than 20% are excluded from the feasible region;
- Seismic risk areas: Main faults and their boundaries are among non-compensatory items in locating problems. The blocks located in seismic risk areas, then, should be removed;
- Flood risk areas: logistics centers play an important role in supply chains and supporting industries, making them vulnerable to floods;
- Environmental boundary of cities: Side effects of logistics center establishments include increased heavy vehicles volume, traffic congestion, air and noise pollution, and traffic safety issues. Therefore, environmental boundary of cities should not be included in the feasible region to avoid the negative impacts of logistics centers;
- Energy lines limits areas: constructions surrounding energy lines (power and gas) are prohibited for safety and security reasons;
- Land use constraints: The establishment of logistics centers on agricultural land would increase infrastructure costs and destroy crops, as well as negatively affect the agriculture sector due to unemployment.

Taking into account the above constraints, feasible region is indicated in Figure 6. In this figure, feasible and infeasible blocks are shown in green and red, respectively. Based on the results, 2883 blocks (41.4%) are feasible which is considered as an input for the next step.

*4.3. Model Outputs*

After excluding the infeasible blocks, the MILP model is run for feasible blocks. At each step the optimal solutions are eliminated from the solution space to extract second-best, third-best, etc., solutions. The value of objective function for total blocks (2883) in descending order is indicated in Figure 7. The highest value of the objective function is estimated, which means the logistics village establishment in the optimal block would

result in IRR 11,140 Billion profit which is almost 3 times the profit of the worst block and 40% higher than the average profit of the blocks. These cost savings can undoubtedly justify the time and cost of location model development. Furthermore, Figure 8 indicates the total profit associated with each block in which blocks with darker color have a higher priority for establishment. In this Figure, the first, second, and third top-ten blocks are highlighted in red, orange, and yellow rectangles, respectively. As shown, the optimal blocks are mostly located in the vicinity of road and rail infrastructures at the east of Qazvin.

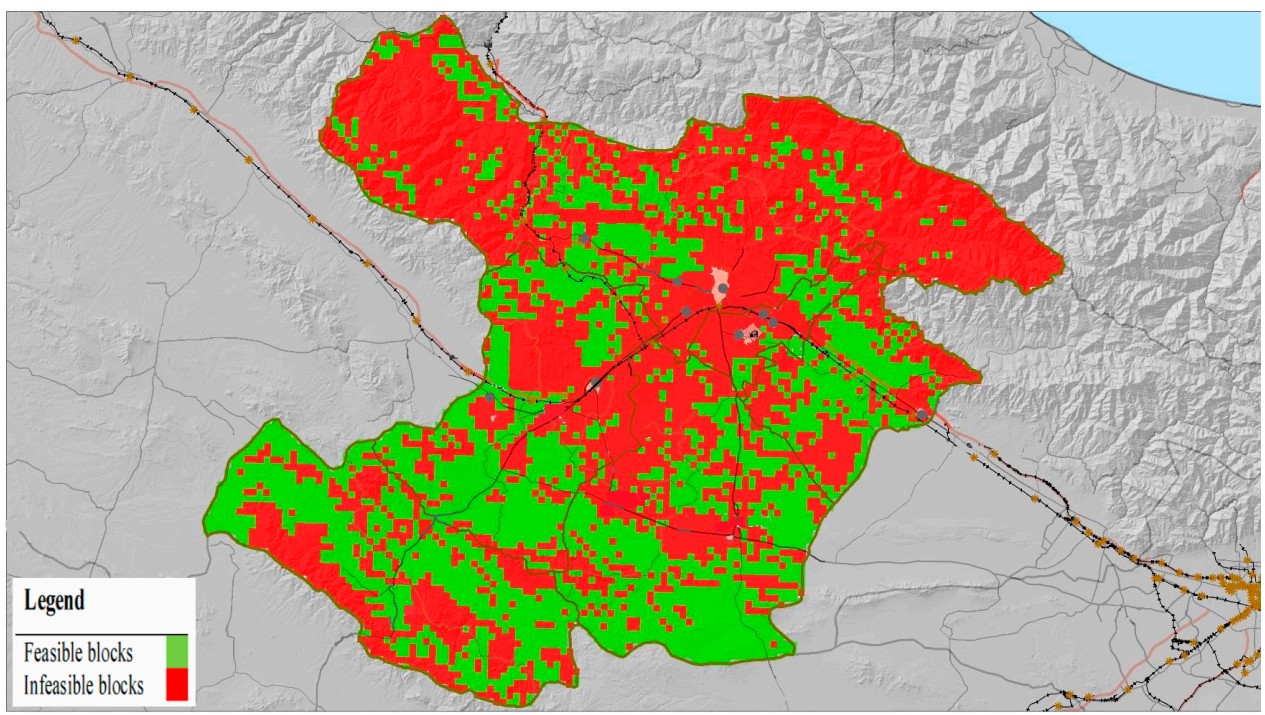

**Figure 6.** Spatial distribution of feasible blocks for logistics village establishment.

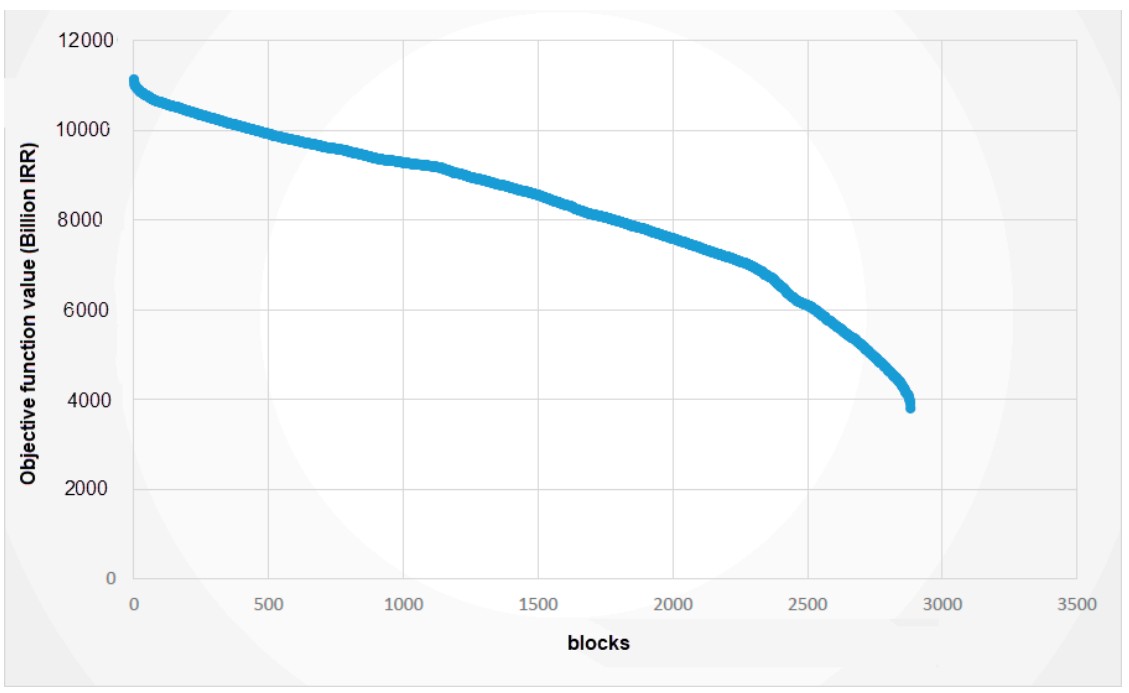

**Figure 7.** Objective function value for each of the 2883 feasible blocks.

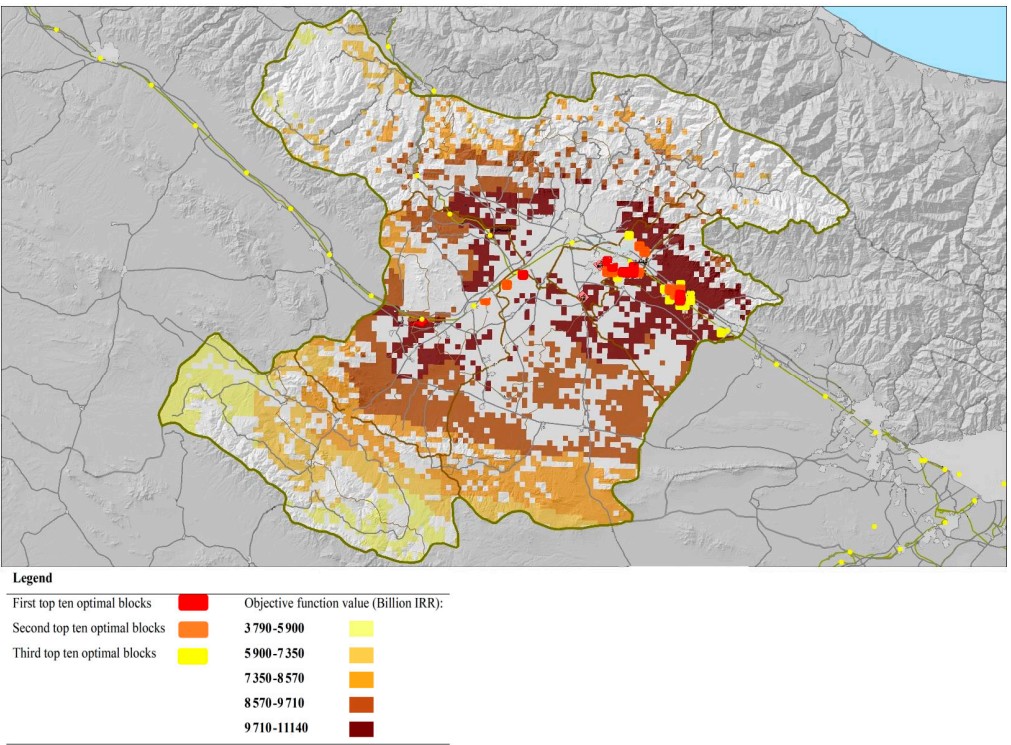

**Legend**

| | | | |
|---|---|---|---|
| First top ten optimal blocks | 🟥 | Objective function value (Billion IRR): | |
| Second top ten optimal blocks | 🟧 | 3 790–5 900 | 🟨 |
| Third top ten optimal blocks | 🟨 | 5 900–7 350 | 🟧 |
| | | 7 350–8 570 | 🟧 |
| | | 8 570–9 710 | 🟫 |
| | | 9 710–11 140 | 🟫 |

**Figure 8.** Classification of blocks based on objective function value.

Figure 9 illustrates a scheme of import/export of Qazvin province by considering the logistics village. As previously discussed in the model, the commodity groups might enter the logistics village if the predicted added value would be higher than the transportation cost. According to the model's output, total freight imported to Qazvin province (7.81 million tons) is directly transferred to their final destinations inside the province without visiting the logistics village. It means that the added value of the logistics village services cannot justify its opportunity cost in terms of transportation cost savings. The reason behind that is the destinations are scattered throughout the province, and smaller in size. Hence, the value added of the logistics village cannot cover its shadow price.

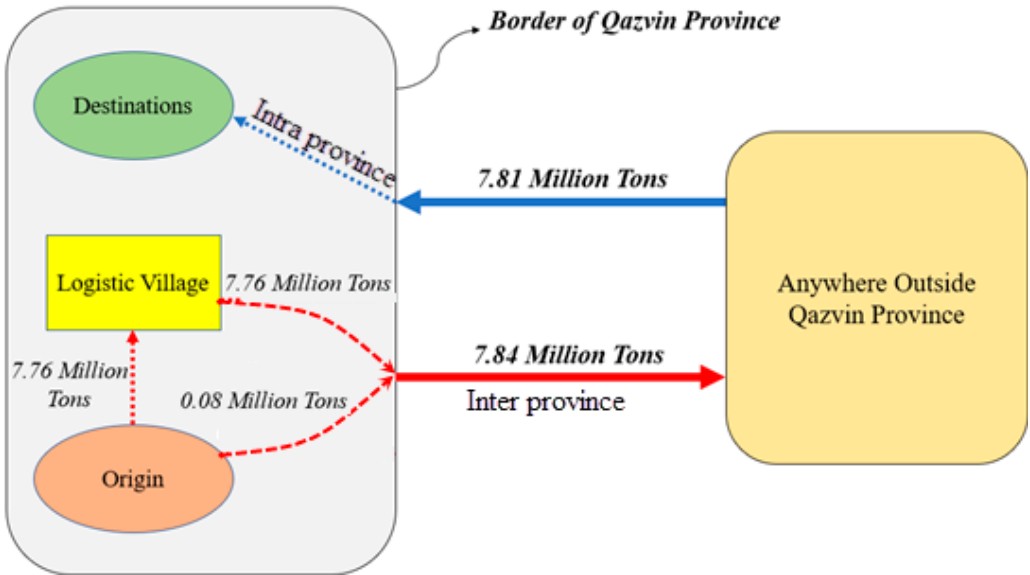

**Figure 9.** Schematic flow of import/export through the logistics village.

The amount of exports of Qazvin province is about 7.84 million tons, of which 0.08 million tons will be delivered to consumers outside the province directly without entering the logistics village. The remaining 7.76 million tons would first enter the logistics village to get a service. This is due to the fact that the export origins are more concentrated, hence the value-added of the services exceed their shadow price (i.e., opportunity cost) in terms of extra transportation and service cost. Additionally, according to the model outputs, the modal share of road and rail for total freight transportation would be 55 and 45%, respectively. Moreover, it is estimated that the cost of import and export between Qazvin and other provinces of Iran would be around IRR 17,320 billion.

## 5. Conclusions

A logistics center is a location where all transportation, logistical, and distribution activities are carried out on a commercial basis. The locating problem of a logistics center as a strategic facility within the logistics networks is a crucial decision in the management of freight transportation networks as well as cost/benefit analysis. Although an extensive body of the logistics literature has focused on location problem, they did not consider various logistics operations such as warehousing, refrigeration, sorting and packaging. The current study aims at filling this gap by proposing a mixed-integer linear programming (MILP) approach and considering these services and their capacities as distinct decision variables. Additionally, the proposed approach allows a trade-off between development costs and operational revenues through the inclusion of investment, operations, and maintenance costs. The proposed model is solved to find a logistics village for a real-sized case of Qazvin province, Iran, as a developing country with a strategic location in international transport corridors.

The study area is divided into 6972 blocks which only 41.4% of the blocks are feasible due to the set of infeasible area including environmental protected lands, land cover constraints, sloped lands, seismic risk areas, flood risk areas, environmental boundary of cities, energy lines limits areas, and land use constraints. The commodity groups are categorized into 6 types including agricultural, industrial, mineral, structural, alimentary, and bestial. We apply CPLEX implementation of branch and cut algorithm using GAMS 24.1.2. The optimal blocks are mostly located in the vicinity of road and rail infrastructures at the east of Qazvin. Based on the results, total freight imported to Qazvin province is directly transferred to their final destinations without visiting the logistics village because of higher transport costs. This is because the cost of transferring freight to the logistics village highly exceeds the profits that can be gained by providing the logistics village's services such as warehousing, refrigeration, sorting, and packaging. Notwithstanding, around 98% of exports of Qazvin province would first enter the logistics village to get a service before delivering to customers. What is more, the modal share of road and rail for total freight transportation would be 55 and 45%, respectively. Government decision makers can use the methodology proposed in this study to analyze and decide the location of these critical facilities.

The research was faced with some limitations, mainly due to the nature of the data and the models used for the analysis. Availability of the data sources such as detailed records of cargo transportation, as well as the resolution of the environmental/natural hazard maps, is the first limitation of the research which forced the researchers to estimate or approximate some of the model's parameters. These estimates and approximations may add an uncertainty component into the model which can be dealt with using stochastic programming or robust optimization approaches in future research. The computational power of the common available hardware is another limitation of the proposed MILP approach for larger problems/problems with higher resolutions. While the optimal solution of the case study was found running simple B&C algorithm on common hardware, future research may investigate the effectiveness of more complicated algorithms including novel cutting planes, hybrid algorithms or even metaheuristic algorithms for larger problems [37,38]. Finally, the environmental aspect of the logistics village in this study was limited to preserving envi-

ronmental protected lands, certain land uses, and land covers, future research can take into account total environmental performance of the logistics village including pollutions due to transportation and operation of the village and provide a trade-off between economic and environmental factors affecting the logistics village location.

**Author Contributions:** Conceptualization, A.B., M.A., S.R. and A.R.M.; methodology, A.B., S.R., A.A. and A.R.M.; software, M.A., S.R. and A.A.; writing—review and editing, A.B., M.A. and S.R.; supervision, A.R.M. and M.S. All authors have read and agreed to the published version of the manuscript.

**Funding:** This research received no external funding.

**Institutional Review Board Statement:** Not applicable.

**Informed Consent Statement:** Not applicable.

**Data Availability Statement:** Not applicable.

**Conflicts of Interest:** The authors have no conflict of interest to declare.

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
