# Peer review of "Logistics Village Location with Capacity Planning Problem, an MILP Model Approach"

_sustainability, doi:10.3390/su15054633_

Round 1

Reviewer 1 Report

Authors have done nice work. However questions asked below could be addressed.

What you give up by making a decision in favor of one thing instead of another is the opportunity cost. How you can connect this with development cost?

In the assumption Multiple cargo types are considered. Need to elaborate what are the multiple cargo types..

Logistics operations such as warehousing, refrigeration, sorting and packaging are considered in this study. What else could be added in this?

"Based on results, total freight imported to Qazvin province is directly transferred to their final destinations without visiting the logistics village because of higher transport cost". Detailed explanations needed.

Author Response

Dear Reviewer

We appreciate your review, the time you have invested, and the invaluable comments which will enable us to improve the manuscript

Reviewer 2 Report

Referee comments

Main Content

This paper focuses on the location and capacity planning problem of a regional logistics center. Different from previous studies on this topic, multiple services are considered in this paper, including transportation, warehousing, refrigeration, sorting and packaging services. Numerical experiment based on Qazvin province was conducted to validate the model and solution method.

The problem investigated is close to the need of an area with large export and import transportation demands. The numerical experiment is based on real data of Qazvin province, which is better than randomly generated cases. However, I still have some questions. Thus, this manuscript should go through a major revision before the final decision can be made.

Major Comments

1.     The part that introduces Mixed Integer Linear Programming (MILP) is not necessary. MILP has been extensively adopted in optimization problems for a long time, and there is no need to give an example.

2.     Although the model proposed considers multiple services, does it bring any essential difference in the model or the solution method? This is an important aspect of the academic contributions of this study.

3.     Since the lifetime costs and revenue are considered, are the transportation demands and other demands for the services provided by the logistics center preset? How are the values of them determined?

4.     I am not sure about the solution method. Is the MILP proposed in this study solved directly by commercial solvers? Then why it is stated that “After excluding the infeasible blocks, the MILP model is run for feasible blocks (2883 times). The value of objective function for each block in descending order is indicated in Figure 7.” This statement, from my point of view, indicates that the optimal solution is obtained by enumerating all the feasible solutions, which is not necessary for a typical MILP model. Then, why choose this solution method.

5.     More detailed analysis on the numerical experiments results should be provided.

Minor comments

1.     There are some format problems in this manuscript.

Page 4, Paragraph 1, this paragraph has smaller font size.

2.     The language can be further polished.

Page 1, Paragraph 1, Line 1–2, “The optimal location for establishing logistics centers is of great importance in reducing logistics costs and supply chain efficiency.” This sentence is kind of ambiguous. Can be rewritten as “…is of great importance in reducing logistics costs and improving supply chain efficiency.”

Author Response

(The authors gave the same response as above.)

Reviewer 3 Report

Congratulations, very interesting study, but I have some questions and comments:

1.       Abstract: The authors should indicate not only the validity of the research but also the gaps in the theory

2.       The authors should clearly distinguish between the terms: logistics center, logistics hub, distribution center, it is important to indicate the differences between these terms. The criteria related to the location of a logistics hub or distribution center do not have to be the same as the criteria for the location of a logistics center.

3.    Why is the location of the logistics center indicated in the title, and the logistics village in the research model? Do the authors treat all these concepts as identical? In literature research, this issue needs to be developed. The sentence in point 3.1 is not sufficient to explain this problem. It is necessary to carry out literature research on these concepts.

4.       Figures 4 and 9 should have a comment

5.       There is no discussion of the results. What new have the authors obtained? How do these results correspond with the results of studies by other authors? What are the limitations of these studies? What are the directions for further research?

6.       The authors referred to the limitations very generally in the conclusions

Author Response

Dear Reviewer

We appreciate your review, the time you have invested, and the invaluable comments which will enable us to improve the manuscript.

Reviewer 4 Report

  1. The authors attempt to formulate an optimization formulation to identify optimal location and capacity for a logistic center using information from the Qazvin province, Iran. Their formulation appears comprehensive and logically correct. 

  1. However, this reviewer has two main concerns. Firstly, while attempting a problem belonging to the general class of facility location problems, the authors do not present a methodological advancement. The formulation appears complete with regard to the interests that the authors consider, but ends up as only an extended version of some existing formulations (for eg., see Tang et al. (2012), Boujelben et al. (2016)). 

  1. Secondly, it is not clear what the authors mean by ‘research conceptual model’ in Section 4. The formulation presented in Section 3 appears to consider multiple simultaneous establishment of logistic centers (or villages). That is, the formulation appears to look for solutions that would prescribe simultaneous setting up of more than one logistics village. This is indicated by the consideration of the subscript ‘i’ (and the summations across i) in the equations. However, the ‘research conceptual model’ appears quite contrary to this. There, the authors treat the problem as a ‘single-hub location problem’. It is not clear why the authors switch from a (seemingly) multiple locations seeking problem to a single-location seeking problem. Are the authors not presenting the solution of the problem presented in Equations 3 - 16? Or are they only presenting a special case thereof? Overall, it appears that the results presented are not in accordance with the optimization problem presented. 

  1. The approach to solve the ‘research conceptual model’ is also not clear. The authors appear to follow a ‘naive’ approach of exhaustively computing the objective function values for all feasible locations and then sorting them to find best locations. 

  1. Internal origin/destination nodes have been referred to as ‘feasible and infeasible nodes’ multiple times in the paper. The authors may consider clarifying the reasons for the same. 

  1. It appears that any Intra-provincial demand and supply (within the internal origin/destination nodes) has been neglected in the formulation (they are also not represented in Figure 3). Wouldn’t this volume of cargo require any of the multiple services that will be offered by the logistic centers? Neglecting such service demands would mean that the solution would be suboptimal in real life.

  1. Equation 7 in page 8, i.e., total cargo receiving service at the logistics village must be lower or equal to total cargo received at the village – is constrained to d ≠ 1. It might be because the constraint balancing in two directions would be redundant; but the authors may consider explicitly stating this.

  1. The ‘M’ used in Equation 9, in page 8, is not explained. It appears that it is an infinitely large number; but please clarify the context.

  1. Are MinG (used in Equation 10) and MinGrid the same? If yes, please make its usage consistent. Also, the inequality mentioned in Equation 10 is ‘≥’; is this correct? Please check..

  1. How would the problem obtain the information on the demand for each service type (warehousing, sorting, packaging)? This has not been clearly explained.

  1. The general form of MILP represented in equation (1), in page 5 (lines 182-186) would be obvious for most readers in this area and therefore appears redundant. 

Minor comments:

  1. In Table 2 of page 7, in the definition for symbol D, ‘from’ has been misspelt as ‘form’. Please correct.

  1. In Table 2 of page 7, in the definition for symbols Dlbm, Dbim and Dbjm, ‘border’ nodes has been misspelt as ‘boarder’ nodes. Please correct.

  1. In line number 259 of page 8, grammatical error in ‘from value added of the services’. Please correct.

  1. In equation 12 of page 8,  fbitdm and Fltdm have been denoted incorrectly as fibtdm and Fbtdm respectively. Please correct.

  1. In line number 273 of page 9, the decision variables are represented incorrectly. Please correct it by using the subscript.

References:

  1. Tang, Lixin, Wei Jiang, and Georgios KD Saharidis. "An improved Benders decomposition algorithm for the logistics facility location problem with capacity expansions." Annals of operations research 210.1 (2013): 165-190.

  2. Boujelben, Mouna Kchaou, Céline Gicquel, and Michel Minoux. "A MILP model and heuristic approach for facility location under multiple operational constraints." Computers & Industrial Engineering 98 (2016): 446-461.

Author Response

Dear Reviewer

We appreciate your review, the time you have invested, and the invaluable comments which will enable us to improve the article.

Round 2

Reviewer 1 Report

Authors have addressed all comments 

Reviewer 2 Report

this version is acceptable.

Reviewer 3 Report

The authors improved the article slightly. It is an article that I can accept, but it is not a good article. I estimate that its scientific value is at an average level.

Reviewer 4 Report

The authors have addressed comments from this reviewer.